# The Expression of Kisspeptins and Matrix Metalloproteinases in Extragenital Endometriosis

**DOI:** 10.3390/biomedicines12010094

**Published:** 2024-01-01

**Authors:** Tatiana Kleimenova, Victoria Polyakova, Natalia Linkova, Anna Drobintseva, Dmitriy Medvedev, Alexander Krasichkov

**Affiliations:** 1Department of Medical Biology, Federal State Budgetary Educational Institution of Higher Education, St. Petersburg State Pediatric Medical University, Ministry of Healthcare of the Russian Federation, 194100 St. Petersburg, Russia; 2Research Laboratory for the Development of Drug Delivery Systems, St. Petersburg State Research Institute of Phthisiopulmonology, Ministry of Healthcare of the Russian Federation, 2-4, Ligovskiy pr., 191036 St. Petersburg, Russia; 3Department of Biogerontology, St. Petersburg Institute of Bioregulation and Gerontology, Dynamo pr., 3, 197110 St. Petersburg, Russia; 4Department of Radio Engineering Systems, Saint Petersburg Electrotechnical University ‘LETI’, 197376 St. Petersburg, Russia

**Keywords:** endometriosis, endometrium cell cultures, KISS1, KISS1R, MMP-2, MMP-9

## Abstract

Endometriosis is characterized by a condition where endometrial tissue grows outside the uterine cavity. The mechanisms of endometrium growth during endometriosis might be similar to the development of a tumor. The kisspeptin (KISS1) gene was initially discovered as a suppressor of metastasis. Matrix metalloproteinases (MMPs) and their inhibitors are described as factors in the early stages of endometriosis and tumor growth progression. We applied the quantitative polymerase chain reaction and the immunofluorescence method to investigate KISS1, its receptor (KISS1R), MMP-2, and MMP-9 in the eutopic and ectopic endometrium in women with and without endometriosis. We presume that the dysregulation of KISS1 and MMPs might contribute to endometriosis pathogenesis. Samples for the immunofluorescence study were collected from patients with a confirmed diagnosis of endometriosis in stages I–IV, aged 23 to 38 years old (*n* = 40). The cell line was derived from the endometrium of patients with extragenital endometriosis (*n* = 7). KISS1 and KISS1R expression are present in the ectopic endometrium of patients with extragenital endometriosis, as opposed to the control group where these proteins were not expressed. There is a decrease in KISS1 and KISS1R values at all stages of endometriosis. MMP-2 and MMP-9 genes express statistically significant increases in stages II, III, and IV of extragenital endometriosis. MMP synthesis increased in the last stages of endometriosis. We suppose that the KISS1/KISS1R system can be used in the future as a suppressive complex to reduce MMP-2 and MMP-9 expression and prevent endometrial cells from invading.

## 1. Introduction

Endometriosis is a chronic disease manifested in a condition where tissue similar to the endometrium of the uterus grows outside the uterine cavity [1]. Endometrial ectopic tissue contains glands and stroma and is capable of functionally reacting to ectogenous, endogenous, or local hormonal stimuli. Endometriosis remains a subject of discussions regarding its etiology, pathogenesis, and treatment. It has been established that endometriosis occurs in one-third of laparoscopies in women with chronic pelvic pains [2]. The expressed clinical picture of endometriosis leading to disability of patients gives the disease not only medical but also social significance [3]. It is responsible for chronic pelvic pain, dyspareunia, dysmenorrhoea, and infertility in women of childbearing age. The pain usually varies during the menstrual cycle, worsening before and during menstrual periods. Menstrual irregularities, such as heavy menstrual bleeding and spotting before menstrual periods, may occur. Misplaced endometrial tissue (ectopic endometrium) responds to the same hormones—estrogen and progesterone (produced by the ovaries)—as normal endometrial tissue in the uterus. Consequently, the misplaced tissue may bleed during menstruation and cause inflammation. The misplaced tissue often causes cramps and pain.

It was previously shown that pregnancy increased the uterine expression of Kiss1 and Kiss1R. These molecules can regulate INFy, MIF, VEGF, IL10, and TNF synthesis. A higher placental expression of Kiss1R mRNA occurred late in the pregnancy, while the expression of Kiss1 was higher in the middle of the pregnancy. We can suggest functional links between Kiss1 and KISS1R in placental angiogenesis and immunology [4]. The hypothalamic–pituitary–gonadal (HPG) axis plays a key role in reproduction, in particular in the synthesis of estrogens. It was shown that kisspeptines play an important role in the regulation of the processes of female follicle development, oocyte maturation, and ovulation through the HPG axis [5].

The overexpression of matrix metalloproteinases (MMPs) contributes to endometriosis development at early stages. Remodeling of the extracellular matrix is a natural occurrence in the normal development, growth, and restoration of tissues. Any change in the balance between the activity of matrix metalloproteinase and their inhibitors (TIMP) can be potentially harmful. It has been suggested that deviating levels of MMP-2, MMP-9, and TIMP could contribute to the development of endometriosis [6,7]. One study has shown an increase in MMP-2 and MMP-9 in the sites of endometriosis compared with the eutopic endometrium [8]. Another study has demonstrated the opposite result: a decrease in MMP-2 in the eutopic endometrium of women with endometriosis [9]. It has been shown that MMPs participate in the reconstruction of extracellular matrix in endometrium implantation sites and promote the penetration of the ectopic focus and its invasion [10]. The level of MMPs, as well as their expression and activity, are controlled by gender hormones [11]. The presence of progesterone and estrogens in the endometrium cell culture decreases metalloproteinase activity, but if hormone levels drop the activity rises steeply, followed by morphological changes in endometrial cells, which are similar to those usingn the uterine epithelial membrane during the menstrual period [12]. Kisspeptin regulates MMPs both at the level of transcription and the level of protein [13,14]. The ability of kisspeptins to suppress MMP-9 transcription to a great extent determines its antimetastatic properties. As we now know, MMP-9 transcription is regulated by AP-1 [15], Sp1 [16], and some other transcription factors, including NFκB [17], through which kisspeptins control the synthesis of MMP-9. Thus, in KISS1-transformed HT-1080 fibrosarcoma cells, researchers found an increased level of IκBα and a significantly reduced quantity of MMP-9 mRNA in comparison with the control [18].

MMP-2 and MMP-9 are the key factors of endometriosis progression and pathogenesis. However, other MMPs also can take part in endometriosis development. MMP-7 promoted epithelial–mesenchymal transition in ovarian endometriosis. MMP-14 regulates the function and formation of invadopodia, which controls the migration and invasion abilities of mesenchymal cells [12].

In our earlier study, it was shown that kisspeptins and their receptors were found in the material organs of the human fetus during intrauterine development [19]. It should be noted that active MMPs can cleave the peptide bond between Gly118–Leu119, binding with KISS-54, the result of which is that three amino acids are cut from the peptide N-end, leading to the inactivation of KISS-54 [20]. This could be a mechanism regulating the feedback between KISS and MMP. There are few studies that examine the role of kisspeptin and its receptor in extragenital endometriosis [21,22,23], and their findings are contradictory. Research into endometrium cell cultures is of interest too, as they could be a convenient object for testing new kisspeptin antagonists. Earlier, we conducted detailed research into invasive and migratory properties of endometrial cell cultures [24]. Produced primarily in the hypothalamus, these neuropeptides regulate pulsatile GnRH secretion and trigger the hypothalamic–pituitary–gonadal axis. Other peripheral organs also express kisspeptin, which inhibits metastasis. Kisspeptin and KISS1R are reported to be present in the endometrium and may play a role in limiting trophoblast migration and invasion into the endometrium during pregnancy to maintain endometrial homeostasis. A deficiency of kisspeptin and KISS1R in the endometrium can lead to pathological conditions, such as endometriosis and endometrial carcinoma. Despite a large enough number of works, such a mechanism still remains insufficiently studied.

The aim of our study was to study KISS1/KISS1R, MMP-2, and MMP-9 gene expression and protein synthesis in endometriosis and compare them with the normal endometrium.

## 2. Materials and Methods

As an object for immunofluorescence and quantitative polymerase chain reaction analysis, we selected samples of endometrial tissue from the uterine cavity and endometrial ectopic tissue on the peritoneum, obtained from the pathomorphology archive of the Research Institute of Obstetrics, Gynecology, and Reproductology named after D. O. Ott. This study involved women with extragenital endometriosis (*n* = 40, age: 23–38 years old). The place of residence of all patients is Saint Petersburg, Russia. This study was conducted according to the guidelines of the Declaration of Helsinki and was approved by the Institutional Review Board (or Ethics Committee) of the Research Institute of Obstetrics, Gynecology, and Reproductology named after D. O. Ott. (Protocol code 88, 8 December 2017). Extragenital endometriosis was diagnosed through laparoscopy with subsequent histology: the stage and phase of the menstrual cycle were confirmed for each eutopic sample of the endometrium. We investigated endometrioid heterotopias on the peritoneum of the pelvis. The material was divided into 5 groups: endometriosis stages I (*n* = 10), II (*n* = 10), III (*n* = 10), and IV (*n* = 10), and the control (*n* = 5). For each sample obtained from patients with endometriosis, 8 histological sections were made. For each sample from the control group, 12 histological sections were made. The calculated value of the samples for this study is less than 350, and larger samples were used for the study. Thus, the sample sizes we estimate are sufficient to ensure that the probability of type 2 error in this study is acceptable. Endometrium samples were collected on the 18th to 22nd days of the menstrual cycle (secretion phase). In the control group, we examined the endometrium and peritoneum tissue samples collected for diagnostic purposes. Since neither extragenital endometriosis nor any other diseases had been found after the histological examination, this material was included in the control group. The cell culture was obtained from 7 patients, and all the material was divided into 2 groups: the control group (*n* = 4, age: 27–38 years old) and the extragenital endometriosis group (*n* = 3, age: 23–35 years old). The endometriosis group included the material from patients with stage I and stage IV endometriosis. In the control group, a biopsy of the endometrium was performed for diagnostic purposes by means of hysteroscopy or laparoscopy and pipel biopsy. The exclusion criteria from the study were an age of over 38 years, women with concomitant diseases of the endocrine and/or reproductive system, and patients whose endometriosis was combined with malignant neoplasms.

This study was approved by the local ethics committee of the Research Institute of Obstetrics, Gynecology, and Reproductology named after D. O. Ott. Informed consent was obtained from all the patients. The tissue subjected to cell culture immediately after removal was placed into a sterile container filled with Dulbecco’s Modified Eagle Medium, supplemented with penicillin/streptomycin, and was transported for in vitro cell culturing. The tissue was next placed on a plastic dish and cut into 1 mm pieces. The minced sample was reconstituted in 2 mL of a digestion solution containing 1 mg/mL of Collagenase II (Gibco, New York, NY, USA, 150–200 units/mg) and incubated for 30 m at 37 °C to dissociate the cells. The reaction was stopped by the addition of the complete culturing medium, and the cells were spun down at 200× *g* for 5 min. The cycle was repeated 5 times. After isolation, the digested tissue was strained in DMEM-F12 supplemented with 10% FBS (Gibco, USA) and 1% penicillin/streptomycin. ICC endometrial cells were cultured on glasses (d = 6 mm; Menzel, Berlin, Germany). Cells were fixed with 4% paraformaldehyde in PBS pH 7.4 for 10 min at room temperature.

We used an enzyme immunoassay to determine the immunophenotypic profile and confirm the mesenchymal origin of the cultured cells. All the obtained endometrial cell lines at the 6th–7th passage had a positive expression of CD9, CD13, CD73, CD90, CD105, CD44, and HLAI. There was no expression of CD31, CD34, CD45, and HLA-DR class II surface markers on the cells. The expression profile of phenotypic markers between different endometrial cell lines was identical. Thus, the obtained endometrial cell lines had an immunophenotypic profile characteristic of mesenchymal cells (CD13, CD44, CD73, CD90, CD105) and not hematopoietic (CD34, CD45, and HLA-DR class II) cells. Based on the obtained data, it can be concluded that an almost homogeneous population of mesenchymal fibroblast-like cells was obtained from the normal human endometrium [25].

### 2.1. Immunofluorescence

The immunofluorescence examination was performed on paraffin sections of 4 μm, which were placed on slides with poly-l-lysine coating. We used the standard single-step protocol with antigen retrieval (high-temperature treatment of tissues) in a 0.01 M citrate buffer (pH = 6.08–6.10) for 20 min at a high pressure and temperature (95–98 °C). Tissues were blocked with 5% BSA protein block (Abcam, Cambridge, UK) for 20 min at room temperature. Proteins were identified with primary antibodies (Table 1).

Cytoplasmic expression was evaluated for KISS1 and cytoplasmic/membranous expression for KISS1R. To differentiate cells in culture, the following antibodies were used: the Anti-E-cadherin antibody and the Anti-Vimentin antibody. Secondary antibodies with fluorochrome Alexa Flour 488 and Alexa Flour 647 (Abcam, UK, 1:1000) were used for visualization. Cell nuclei were stained with DAPI (AppliChem, Darmstadt, Germany) for 1 min. The ready preparations were placed under slides mounted in Fluorescent Mounting Medium (Dako, Glostrup, Denmark). The primary antibody was omitted for negative controls. Visualization of the samples was performed with a FlueView 1000 (Olympus, Tokyo, Japan) confocal laser scanning microscope. Verification of protein expression was performed using lasers with wavelengths of 650 nm, 500 nm, and 405 nm for the visualization of cell nuclei. Five fields of view were examined with ×40 magnification.

To evaluate the results of immunofluorescence staining, a morphometric study was conducted using the microscope image analysis system, an IntelPentium 5 personal computer, and Image J version 1.52u (the developer—Wayne Rasband, National Institutes of Health, USA, Rockville Pike, Bethesda, MD, USA) software. Five fields of view were analyzed in each case. The expression area was calculated as a percentage of the area of immunopositive cells and the total area of cells in the field of view. Quantifying the area of expression is more accurate than counting immunopositive cells in the field of view [26,27]. The process of analyzing microphotographs was carried out according to the following algorithm. 1. Open a microphotograph in the ImageJ program. 2. Set the pixel/micron ratio. 3. Convert the image to RGB stack (the program only works with black and white images). 4. Select the studied parameters: expression area, average value, minimum, and maximum. 5. Tissue detection and discard of glandular lumen. Points 4 and 5 were performed by use of a freehand selection tool measuring immunopositive pixels and the area of expression of the investigated markers. A similar approach was used by Donnez O. et al. in their research [28]. 6. Use Threshold to apply masks (examples of microphotographs with masks are shown in Figure 1). 7. Save the data in an Excel table (at this stage, it is necessary to manually delete artifacts if they are on microphotographs).

### 2.2. RNA Extraction and Quantitative Polymerase Chain Reaction

Total RNA from the eutopic sample of the endometrium and normal endometrium was stabilized using an IntactRNA RNA stabilization solution (Evrogen, Moscow, Russia). RNA isolation was performed using an RNeasy MiniKit (Qiagen, Hilden, Germany, FRG) in accordance with the recommendations of the manufacturer. Spectrophotometric analysis was used to assess the purity of RNA. The ratio of optical density at wavelengths of 260 and 280 nm (A 260/280) was about 2. The integrity of each RNA sample was estimated on the basis of fragment size distribution indicated by two peaks corresponding to 18S and 28S ribosomal RNAs and a signal from small RNAs. The quality of RNA was assessed based on RNA integrity number (RIN) values ranging from 1 to 10, with 1 being the most degraded and 10 being the most intact, using an RIN algorithm [29]. The RNA integrity number (RIN) in our investigation was 8. This is acceptable for data analysis [30]. The concentration range of isolated RNA was near 800–1000 ng/μL. The first strand of cDNA was synthesized via a Revert Aid First Strand cDNA Synthesis Kit (Thermo Fisher Scientific Inc., Waltham, MA, USA) using 100 ng of RNA per 20 µL of the reaction mixture. The obtained cDNA was used directly as a template for quantitative PCR at 1 µL per 24 µL of the reaction mixture. Quantitative PCR was performed by means of a DT-322 (DNK-Technology, Moscow, Russia) using a qPCRmix-HS SYBR + ROX amplification kit (Evrogen, Russia). Quantitative PCR was used to quantify the expression of *KISS1*, *KISS1R*, *MMP-2,* and *MMP-9* genes. Oligonucleotide primers were designed using the NCBI Primer-Blast online service. Primer pairs, in which one of the primers corresponded to regions of two adjacent exons, were used. The synthesis of oligonucleotides was carried out at NPO Syntol (Moscow, Russia). Primer sequences for *KISS1*, *KISS1R*, *MMP-2,* and *MMP-9* genes are presented in Table 2. The expression level relative to the reference *GAPDH* housekeeping gene was determined by the 2-ΔΔC_q_ method. Various genes, including *GAPDH*, are used as housekeeping genes during PCR analysis for endometrial cells [31,32]. Statistical processing of the results and plotting of the diagrams was carried out via “Microsoft Excel 2010”. Three independent samples from each group (biological parallels) were used in the research. For each cDNA sample, a minimum of three parallel reactions in adjacent slots (technical parallels) were performed.

### 2.3. Scratch Testing

A scratch test was performed with a high µ-Dish at 35 mm (Ibidi). The culture was planted in two wells at concentrations of 35,000 cells/mL, the monolayer was brought to a confluence of 90–95%, and a “wound” was made on the monolayer by removing a special frame from the Petri dish. The detached cells were removed by DPBS washing, and a fresh medium was added. Scanning was performed in 12, 24, 48, and 72 h on an Olympus CKX53 inverted microscope with the integrated phase contrast system at 40×.

### 2.4. Cell Invasion Assays

To evaluate cell invasion, we used Falcon inserts in 24-cell trays (BD Biosciences, Franklin, NJ, USA) with pore sizes of 8 μm. The endometrial cell suspension in the DMEM/F-12 medium was dispensed into the upper chamber (insert), while FCS (10%) was transferred to the lower chamber (well). In the second series of experiments, autologous peritoneal fluid was added to the lower chamber. The trays were incubated for 10 h at 37 °C in the atmosphere with 5% CO_2_, and then the number of cells was calculated, which penetrated through insert pores into the lower chamber.

### 2.5. Analysis and Statistics

IHC and ICC involve using ImageJ (NIH). The obtained data were statistically processed using STATISTICA version 10.0 software (Statsoft Inc., Tulsa, OK, USA). A data normality test was performed using the Shapiro–Wilk criterion, and homogeneity of variances was evaluated with the Levene criterion. Medians (25–75 percentiles) were used when there was no normal distribution of the sample. Values of *p* < 0.05 were considered statistically significant.

## 3. Results

### 3.1. Immunohistochemistry

According to the analysis of the unit area of the KISS1 expression in the eutopic endometrium, statistically significant differences were found in the comparison of the control group with stage I (*p* = 0.008) and stage III (*p* = 0.03) of extragenital endometriosis (Figure 2a). The examination of the kisspeptin receptor revealed statistically significant differences in the comparison of the control group in stage I (*p* = 0.003), stage III (*p* = 0.02), and stage IV (*p* = 0.03) of extragenital endometriosis (Figure 2b). We observed the greatest KISS1/KISS1R expression in the control group (Figure 2a,b). The study results provide evidence that kisspeptin and its receptor are present in endometrial ectopic tissue as opposed to the control group, where such proteins were not expressed (Figure 2c). The maximum values of KISS1/KISS1R were recorded at stage IV of endometriosis.

The micrographs in Figure 1 show the results of the immunohistochemical study. The expression of KISS1/KISS1R was observed in the glandular component of the endometrium; no reaction to these proteins was found in the stroma (Figure 2a). In the case of endometrial ectopic tissue, a reaction to KISS1 and KISS1R proteins was observed in ectopic endometrial glands. It was shown that the area of MMP-9 expression in endometrial ectopic tissues has a tendency to increase in all stages of extragenital endometriosis (Figure 3a–c). The analysis of the MMP-9 area expression in the glandular component of the endometrium revealed a statistically significant increase (*p* < 0.05) in comparison with the control group at all stages of endometriosis (*p* = 0.002; *p* = 0.002; *p* = 0.002; *p* = 0.003 for stages I, II, III, and IV, respectively) (Figure 3b). It was also shown that there was a statistically significant increase in MMP-9 area expression in the stroma of the endometrium in the case of extragenital endometriosis (*p* < 0.05) in comparison with the control group at all stages of endometriosis (*p* = 0.004; *p* = 0.006; *p* = 0.003; *p* = 0.003 for stages I, II, III, and IV, respectively) (Figure 3a). It was found that MMP-2 expression in endometrial heterotopias and the endometrium increased at all stages of endometriosis (Figure 3d–f). The relative area of MMP-2 expression in glands was a statistically significant difference (*p* < 0.05) during stage IV of extragenital endometriosis in comparison with the control (*p* = 0.021) (Figure 3f). At the same time, the control had the lowest rates of all groups, but significant statistical differences were found in stage IV.

### 3.2. KISS1, KISS1R, MMP-2, and MMP-9 mRNA Gene Expressions in Ectopic and Normal Endometriums

The relative level of *KISS1* mRNA expression showed a statistically significant decrease in stages II and III of extragenital endometriosis; they were 3.5 and 1.9 times lower, respectively, compared with the control. The *KISS1R* mRNA relative level of expression statistically significantly decreased in the II, III, and IV stages of extragenital endometriosis by 4.9, 2.5, and 2.2 times in comparison with the control. The relative level of MMP-9 mRNA expression increased statistically significantly by 3.2, 3.4, and 4.3-fold in the II, III, and IV stages of extragenital endometriosis, respectively, compared with the control (Table 3). Similarly, the relative level of MMP-2 mRNA expression increased statistically significantly by 2.9, 4.1, and 4.3-fold in the II, III, and IV stages of extragenital endometriosis, respectively, compared with the control (Table 3). Similarly, the relative level of MMP-2 mRNA expression increased statistically significantly by 2.9, 4.1, and 4.3-fold in the II, III, and IV stages of extragenital endometriosis, respectively, compared with the control (Table 3).

A decrease in the *KISS1* and *KISS1R* gene expressions and an increase in the MMR-2 and MMR-9 gene expressions in endometriosis heterotopias compared with a normal endometrium correlates with the changes in the synthesis of the corresponding proteins detected by the immunohistochemistry method.

### 3.3. Scratch Testing

The primary cell culture of the endometrium is endometrial ectopic tissue and contains cells with fibroblast-like (eSF) and epithelium-like (eEC) morphology. After 24 h of cultivation, cells began to migrate from non-dissociated conglomerates, forming colonies of small round cells with epithelium-like morphology. Cells with typically fibroblast-like morphology were also observed.

Then, the fibroblast-like cells proliferated and formed colonies, while in epithelium-like colonies, some cells died within 72 h. Immunofluorescence analysis allowed us to determine that eEC cells express E-cadherin, a membrane protein of epithelial origin. The number of glandular cells varied from sample to sample (S_exp.E-cadh_ = 5.72–38.29%) and was not stable, which precludes using this parameter as a key one for estimating the number of glandular cells in the resulting culture. Vimentin (a cytoskeletal marker of stromal cells) was used as a marker for eSF cells. Cell lines were essentially different in their migratory capacity. In the control samples of cell cultures after 24 h, the “wound” area was 27.29 ± 2.61% of the specimen area, and after 48 h, 17.61 ± 1.63%. In the culture of extragenital endometriosis patients, the wound area was 20.77 ± 2.59% after 24 h and 8.21 ± 1.94% after 48 h (*p* = 0.003) (Figure 4).

### 3.4. Cell Invasion Assays

In this study, we conducted an analysis of cell culture invasion based on measuring the motility of cells and cell movement in a gradient of chemoattractant (one fetal calf serum, two autologous peritoneal fluids). In the control group, the number of cells that passed through the transaerobic filter (pores of 8 μm) and attached to the surface varies from 4.6 to 16.3 in five fields of view (Figure 5a). In the cell cultures obtained from patients with endometriosis, significant differences were not observed, with 6.5–12.8 cells per field of view (Figure 5b). Adding peritoneal fluid into the lower chamber did not affect the migratory capacity of cells in the control group but increased invasion in the experimental group.

### 3.5. Immunocytochemistry

It was demonstrated that KISS/KISS1R was present in individual cells in the primary cell culture. This study allowed us to determine that the unit area of kisspeptin expression tended to decrease to the level of the protein under investigation in cases of extragenital endometriosis. The lowest KISS1 figures were observed in patients with stage IV endometriosis (S_KISS 1_= 0.568 ± 0.133%) (Figure 6a). In the control value, S_KISS1_ = 1.599 ± 0.474%. KISS1R was the same in the control (S_KISS1R_ = 2.327 ± 1.174%) and disease (S_KISS1R_ = 2.456 ± 1.029%) (Figure 6b).

This study demonstrated that MMP-2 and -9 were present in individual cells in the primary cell culture. The analysis of MMP-2 and -9 in cell culture showed no statistically significant differences between the control group (S_MMP-2_ = 2.362 ± 0.446%; S_MMP-9_ = 1.832 ± 0.526%) and experimental group (S_MMP-2_ = 2.155 ± 0.962%; S_MMP-9_ = 3.831 ± 4.299%) (*p* > 0.05) (Figure 6c). A sharp spike in matrix metalloproteinase 9 was observed only in the cell culture from a single extragenital endometriosis patient (S_MMP-9_ = 8.789 ± 1.927%), while in other patients in that group, no such changes in the MMP-9 expression level were registered (Figure 6d).

## 4. Discussion

Based on our data, it can be assumed that kisspeptin and its receptor were produced only by endometrial epithelial cells. The Stroma of the endometrium is a connective tissue that provides the strength of the mucous membrane and connects the cells of the endometrium. In the basal layer, the stroma is dense and consists of connective cells and a large number of thin collagen fibers. Since the stroma does not contain epithelial cells, the expression of kisspeptin and its receptor was not detected.

It was shown that the KISS, KISS1R, MMP2, and MMP9 mRNA levels in ectopic and normal endometrium do not always define directly corresponding protein synthesis. mRNA expression is not always directly related to the amount of protein synthesized and its functional activity. After the synthesis of the primary amino acid sequence of the protein or, more often, its precursor, folding occurs. Violation of folding can lead to a lack of synthesis of the functional structure of the protein and, in this case, despite the high level of mRNA expression, protein synthesis will be insignificant.

The findings obtained can provide evidence of the role of kisspeptin in certain groups of patients with endometriosis. The presence of kisspeptin in endometrial ectopic tissue might also suggest that kisspeptin as a metastasis-suppressing protein may act as a constraining factor. Active forms of MMP-2 and MMP-9 are particularly important at early stages of extragenital endometriosis development. Any changes in the balance between the activity of MMPs and their inhibitors can contribute to endometriosis development. Increased MMP-2 expression in the ectopic endometrium can indicate aggressive progression of endometriosis. KISS1 is a regulator of MMPs and is capable of suppressing protein transcription. Thus, we may suppose that MMP-2 and MMP-9 are less suppressed in endometriosis patients, which, in turn, can lead to active migration (cell invasion). Kisspeptin antagonists could be used for the treatment of women in postmenopause, hyperovarianism, polycystic ovary syndrome, endometriosis, and uterine myoma. Complete suppression of luteinizing hormone (LH) by GnRH analogs may lead to side effects, such as increased temperature, loss of libido, and decreased bone mineral density [33]. While the complete suppression of gonadotropins, estrogens, and progesterones is necessary in the case of prostate cancer, incomplete hormone suppression seems more reasonable for the treatment of endometriosis. Clinical benefits in the treatment of endometriosis and uterine myoma with the suppression of GnRH and the use of selective progesterone receptor modulators [34] allow us to suggest that approaches that are not based on complete suppression of the GnRH axis have clear clinical significance. The targeted incomplete suppression of gonadotropin by means of kisspeptin has the potential to eliminate the existing limitations of GnRH analogs, although new evidence on KISS1 analogs demonstrates their potential for the complete suppression of hormones [35]. Our findings are consistent with the hypothesis that a lower expression of KISS1/KISS1R contributes to the pathogenesis of endometriosis by increasing invasion of tissues. In particular, a decrease in KISS1 signal transmission may weaken the suppression of MMP-2 and MMP-9, thereby allowing for an increase in the degradation of the extracellular matrix. This is consistent with findings that the MMP-2 and MMP-9 activity is increased in women with the eutopic endometrium compared to endometriosis. Such increased MMP activity is considered to contribute to the pathogenesis of endometriosis. It has been established that MMPs play an important role in the development of endometrial ectopic tissue. Our study has demonstrated that KISS1, KISS1R, MMP-2, and MMP-9 are present in the ectopic endometrium. MMP-9 is known to have a direct effect on endometriosis sites through the system of cytokines and growth factors, affecting angiogenesis and adhesive processes. Our findings have shown the unit area of MMP-2 expression increases in the glandular component of the endometrium with endometriosis in comparison to the control group, which may indicate an increased activity of MMP-2 in the case of pathology and is consistent with the literature data on the increased activity of matrixins in extragenital endometriosis development [36,37]. Moreover, we observed an increased trend toward a change in the unit area of MMP-2 expression in the glandular component of the endometrium depending on the stage of extragenital endometriosis. On the one hand, this might be evidence of the increased proteolytic activity of MMP-2 in pathology development and, consequently, increased retrograde menstruation with an increase in the speed of endometrium ejection. In this case, the increased expression of MMP-2 could be a manifestation of molecular cascades of positive induction which develop in progressing pathology. On the other hand, one cannot exclude the presumed suppressor activity of MMP-2 with regard to angiogenesis. In that case, the increase in MMP-2 expression might be a response aimed at restoring the normal microflora. Therefore, research should be continued to find signaling molecules related to MMP-2 activity. It should be also taken into account that MMP-9 and MMP-2 may be involved in the processing of certain molecules that can have a significant impact on the development and progression of pathology. Our results regarding an increase in the optical thickness of MMP-2 expression in the glandular component of the endometrium with extragenital endometriosis of stage I and stage IV, in comparison with the control group, give evidence of an increased activity of this protein in the case of pathology, which is consistent with the literature data [38,39]. Since MMP-2 and MMP-9 can break down and inactivate KISS1, an increase of MMP activity in endometriosis can work through feedback to decrease KISS1 levels, additionally eliminating possible inhibiting effects of KISS1 on migration and invasion.

Using the scratch test method, we found statistically significant differences that may be related to the ability of kisspeptin to inhibit cell migration. In endometriosis, cell migration occurs faster than in normal endometrium.

The obtained results can be explained by an increase (in the peritoneal fluid of extragenital endometriosis patients) in the number of cytokines activating angiogenesis, such as IL-1β, IL-6, and IL-8 [40], and stimulating the adhesion of endometrial cells to the peritoneal mesothelium TNFα [41], as well as vascular endothelial growth factors (VEGFs) [42], which enhance the invasive properties of cells in culture. The kisspeptin receptor expression level did not change over the course of our study. It has been found that in the cell culture of endometriosis patients, the KISS1 level was lower, and the migration activity of this line was statistically increased in comparison with the control group. It should be noted, however, that these differences might be influenced by individual characteristics of patients from whom the endometrial material was obtained: their endocrine profile, age, peculiarities of endometriosis progression, concurrent diseases, etc. The literature reviews describe research that has found that the activation of KISS1R by KISS1 inhibits cell mobility, including proliferation, invasion, chemiotaxis, and metastasis formation [43,44,45,46]. Immunofluorescence analysis allows for the identification of two or more proteins at a time, which helps to track the simultaneous presence of proteins under study. This study has demonstrated the expression of kisspeptins and receptors to kisspeptins with a marker protein of epithelium-like endometrial cells in the same cells, which is evidence that only eECs are capable of expressing KISS1/KISS1R.

Many studies have shown that MMPs promote tumor invasion and metastasis throughout the body [47,48]. In some studies, it was found that in endometrial cancer, a high expression of MMP-2 and a low expression of TIMP-2 are important tumor markers [49]. In our study, it was shown that MMP-2 and -9 are in the primary cell culture of the endometrium in single cells. Analysis of the expression of MMP-2 and -9 in culture showed that there were no statistical differences between the control and the study group (*p* > 0.05); we assume that the sample was insufficient. Our data are inconsistent with the data of other studies, which showed an increase in MMPs in the endometrial adenocarcinoma cell line [50], in serous uterine cancer [51], and in endometrioid uterine cancer [33,52]. Kisspeptin may modulate embryo implantation and decidual programming in human pregnancy. The extravillous trophoblast invades the maternal decidua during embryo implantation and placentation. The motile behavior and invasive potential of decidual stromal cells regulate embryo implantation and programming of human pregnancy. Kisspeptin agonists can reduce cell motility. Kisspeptin agonists can reduce cell motility and suppress MMPs: MMP-2 (decreased by 36%) and MMP-9 (decreased by 23%) [53]. Therefore, we may presume that KISS1, KISS1R, MMP-2, and MMP-9 can be used for the diagnostic, assessment of the progression of endometriosis and the effectiveness of treatment of this disease in women from 23 to 38 years old without endocrine system pathology and oncology diseases. The application of these signal molecules can be used for endometriosis diagnosis in women over 38 years old. Patients with endocrine or oncology pathology require more detailed research because it was not included in the objectives of this investigation.

## 5. Conclusions

Kisspeptin is an important regulator of the HPG axis and a target for a number of proteins, such as steroid hormones, and participates in the normalization of nutrition and metabolism. Binding with KISS1R in HPG axis neurons, kisspeptin activates regulators which, in turn, provide a mandatory signal for HPG axis secretion, thereby blocking the cell cascades that control fertility. We suppose that the KISS1/KISS1R system can be used in the future as a suppressing complex to reduce MMP expression and prevent endometrial cells from invagination. Understanding the kisspeptin role can lead to using it as a biomarker in infertility treatment. As far as we are currently aware, kisspeptin analogs have not yet been studied with regard to the treatment of endometriosis. Therefore, we may presume that kisspeptin suppression might play a certain role in extragenital endometriosis pathogenesis and progression.

## Figures and Tables

**Figure 1 biomedicines-12-00094-f001:**
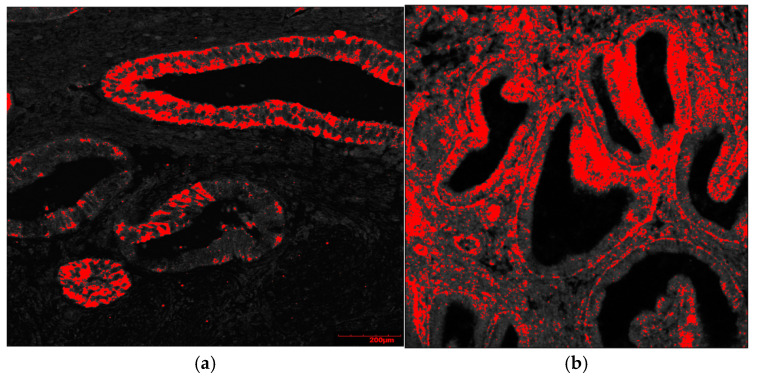
(**a**) Masks corresponding to KISS1 expression in an eutopic endometrium in the control group, (**b**) masks corresponding to MMP-9 expression in an eutopic endometrium in patients with extragenital endometriosis.

**Figure 2 biomedicines-12-00094-f002:**
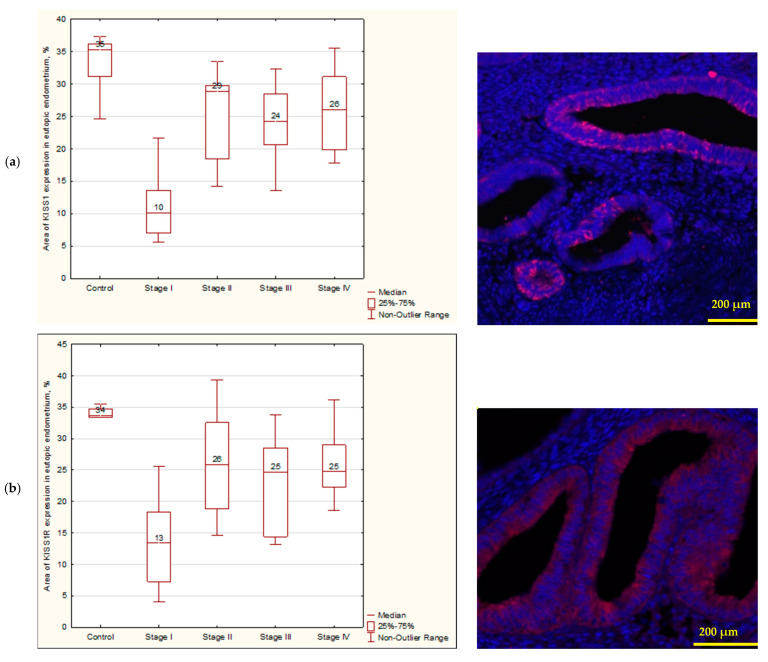
(**a**) KISS1 in an eutopic endometrium in the control group and patients with extragenital endometriosis and KISS1 expression in an eutopic endometrium; statistically significant differences were found in the comparison of the control group with stage I (*p* = 0.008) and stage III (*p* = 0.03) of extragenital endometriosis; (**b**) KISS1R in an eutopic endometrium in the control group and patients with extragenital endometriosis. The examination of the kisspeptin receptor revealed statistically significant differences in the comparison of the control group with stage I (*p* = 0.003), stage III (*p* = 0.02), and stage IV (*p* = 0.03) of extragenital endometriosis.; (**c**) KISS1 in an ectopic endometrium in the control group and patients with extragenital endometriosis; (**d**) KISS1R in an ectopic endometrium in the control group; nuclei stained with DAPI; magnification ×40. Microphotographs in this figure illustrate a general appearance of the investigated tissue.

**Figure 3 biomedicines-12-00094-f003:**
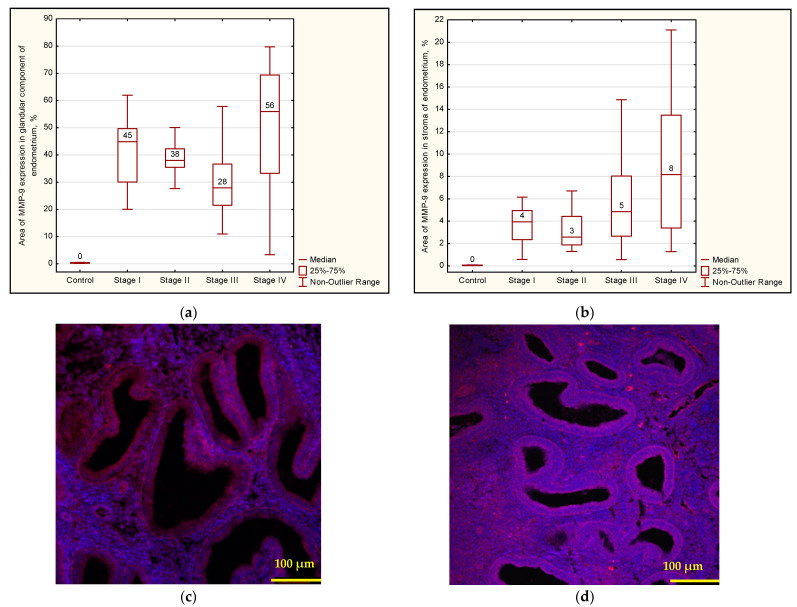
(**a**) MMP-9 in the glandular component of the endometrium; (**b**) MMP-9 in the stromal component of the endometrium; 0—control, 1—minimal (stage I), 2—mild (stage II), 3—moderate (stage III), severe 4—(stage IV); (**c**) MMP-9 in an eutopic endometrium in patients with extragenital endometriosis; (**d**) MMP-2 in the stromal component of the endometrium; (**e**) MMP-2 in the glandular component of the endometrium; (**f**) MMP-2 in an eutopic endometrium in patients with extragenital endometriosis; nuclei stained with DAPI; magnification ×20.

**Figure 4 biomedicines-12-00094-f004:**
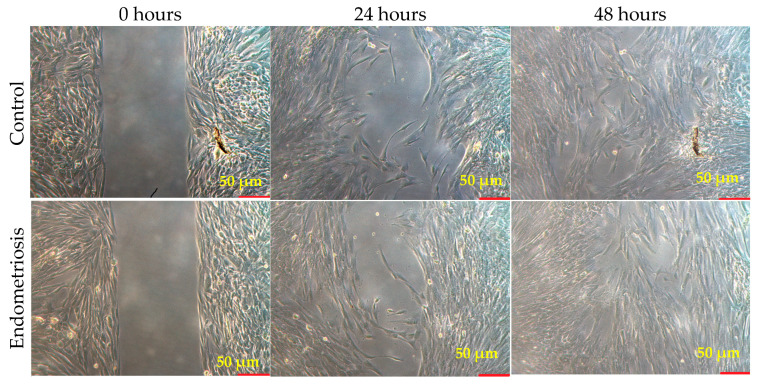
Scratch test; magnification ×40.

**Figure 5 biomedicines-12-00094-f005:**
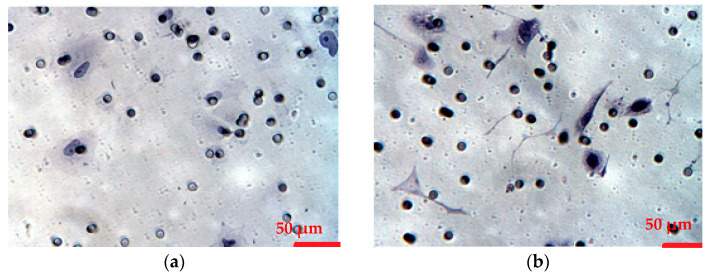
(**a**) Control cell culture; (**b**) endometriosis cell culture; magnification ×40. In the control group, the number of cells that passed through the transaerobic filter and attached to the surface varies from 4.6 to 16.3 in five fields of view. In the cell cultures obtained from patients with endometriosis, significant differences were not observed, with 6.5–12.8 cells per field of view.

**Figure 6 biomedicines-12-00094-f006:**
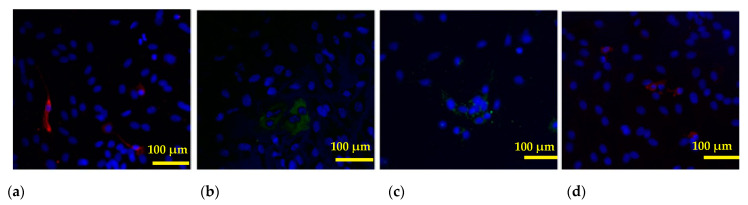
Proteins in the endometrial cell culture in the case of extragenital endometriosis: (**a**) KISS1; (**b**) KISS1R; (**c**) the active form of MMP-2; (**d**) the active form of MMP-9; KISS1 and MMP-9—red fluorescence (Alexa Flour 647); KISS1R and MMP-2—green fluorescence (Alexa Flour 488); nuclei stained with DAPI; magnification ×40.

**Table 1 biomedicines-12-00094-t001:** Primary antibody characteristics and incubation times.

Antibody, Animal Host Corresponding to the Clone	Firm, Clone	Dilution	Concentration	The Time of Incubation	Reactivity
Anti-Kisspeptin 1 (KISS1), mouse monoclonal	abcam, EPR23770-189,	1:1000	100 µg at 0.5 mg/mL	60 min at room temperature	Human
Anti-KISS1R, rabbit polyclonal	abcam, ab221859	1:200	100 µg at 0.5 mg/mL	60 min at room temperature	Mouse, Human
Anti-MMP-2, mouse monoclonal	abcam, 6E3F8	1:200	100 µg at 1 mg/mL	30 min at room temperature	Mouse, Rat, Human
Anti-MMP-9, rabbit monoclonal	abcam, EP1254	1:150	100 µg at 1 mg/mL	60 min at room temperature	Rat, Human, Recombinant Fragment
Anti-E-cadherin, mouse monoclonal	Novus Biologicals, NBP2-19051	1:50	100 µg at 1 mg/mL	15 min at room temperature	Human, Mouse, Monkey
Anti-Vimentin mouse monoclonal	Dako, M0725	1:400		60 min at room temperature	Black Ferret, African Green Monkey, Human, Mouse, Rat, Bovine, Swine, Domestic Sheep, Domestic Rabbit

**Table 2 biomedicines-12-00094-t002:** Primer sequences for *KISS1*, *KISS1R*, *MMP-2,* and *MMP-9* genes.

Gene	Sequence
*KISS1*	Forward 5′-TCGCTGGTCATCTACGTCTGC-3′
	Reverse 5′-GCTGGATGTAGTTGACGAACTTCG-3′
*KISS1R*	Forward 5′-AACTCACTGGTTTCTTGGCAGCTA-3′
	Reverse 5′-AGGAGTTCCAGTGTAGTTCGGCA-3′
*MMP-2*	Forward 5′-CCATCGAAGCAAAGGTGACAACCGTGA-3′
	Reverse 5′-GGACTAGTGGCTGGAAGAGTGCTGGC-3′
*MMP-9*	Forward 5′-CCATCGATTAGAAGCAGGAGGACCCGA-3′
	Reverse 5′-GGACTAGTTGGCTAACGCTGCTTTG-3′

**Table 3 biomedicines-12-00094-t003:** *KISS1, KISS1R, MMP-2,* and *MMP-9* mRNA gene expressions in an ectopic endometrium heterotopy and a normal endometrium (control).

Group	Relative mRNA Expression			
	*KISS1*	*KISS1R*	*MMP-9*	*MMP-2*
Control	1.27 ± 0.19	1.93 ± 0.20	0.33 ± 0.03	0.27 ± 0.04
Endometriosis stage I	1.02 ± 0.13	1.80 ± 0.17	0.40 ± 0.06	0.35 ± 0.05
Endometriosis stage II	0.36 ± 0.05 *	0.40 ± 0.05 *	1.07 ± 0.11 *	0.79 ± 0.06 *
Endometriosis stage III	0.68 ± 0.09 *	0.76 ± 0.09 *	1.12 ± 0.14 *	1.11 ± 0.12 *
Endometriosis stage IV	0.88 ± 0.12	0.89 ± 0.12 *	1.41 ± 0.20 *	1.17 ± 0.10 *

*—*p* < 0.05 in comparison with the corresponding control.

## Data Availability

Data is unavailable due to privacy or ethical restrictions.

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
