# Peer review of "The Expression of Kisspeptins and Matrix Metalloproteinases in Extragenital Endometriosis"

_biomedicines, 2024, doi:10.3390/biomedicines12010094_

Round 1
Reviewer 1 Report
Comments and Suggestions for Authors
Brief report: The manuscript, titled "Biomedicine-2662133," aims to investigate the expression of KISS1/KISS1R, MMP-2, and MMP-9 in endometriosis compared to healthy endometrium. The aim is clearly stated, although modest. The study design is adapted to the aim. Specific methods must be addressed. The statistics must be improved. The results are coherent, their presentation must be improved. The discussion is speculative. The conclusion is partially in line with the findings.
Detailed report:
Abstract:
The abstract should be revised to provide more informative results. The grammar must be improved. In particular, the abstract does not define or introduce MMPs. A conclusive remark should be added.
Introduction:
The presentation of clinical manifestations is not directly relevant to the study.
Please explain the spatio-temporal expression of kisspeptins in the uterus, particularly in the endometrium, and discuss the role of oestrogens in kisspeptin regulation.
The specific actions of the MMPs studied, including their substrates and regulation. The authors should consider discussing the potential involvement of other MMPs, including transmembrane forms.
The process of collective cell migration is not addressed; in fact, the referee specifically recommends resubmitting after investigating this key element of endometriosis in regard to kisspeptins.
Materials and Methods:
L95: A table with demographics should be added. Please indicate all relevant parameters.
L102: inclusion and exclusion criteria of the patients are not given.
L139: replace « immunohistochemistry » by « immunofluorescence » as the authors used indirect fluorescence and no chemical substrate (DAB, TSA,..).
Table1: Indicate the animal host corresponding to the clone. Indicate the concentration of antibody (as µg/ml)
L162: The referee asks the authors to specify whether the quantifications were performed excluding the lumens, and if so, from what size onwards. This part must be widely extended: the authors report « stroma » vs « glandular » ROIs in Fig2. How these ROIs were defined? Was some signal normalisation performed? The authors are encouraged to provide technical illustrations of the masks to demonstrate the methodological validity. In particular, for MMPs: could the authors measure stroma expression gradients around existing glands?
L164: The reviewer strongly disagrees with the statement of superior accuracy of the Digital image analysis method by relative marker area. Ref 24 does not support the statement, and reference 25 refers to fluorescence intensity.
L169: Please indicate if the purity was assessed (A260/A280 nm).
170: Replace « quality » by « integrity ».
L172: The range of obtained RIN values is not given. Please note that RIN range covers 0-10 values. For gene expression study, please note that material with RIN lower than 7 is usually not acceptable.
L185: the authors use GAPDH as HKG but this is not justified. Did the authors previously demonstrate the validity of this HKG? Give a reference or demonstrate validity.
L186: please amend as « 2-ΔΔCq ».
L198: indicate the reference of the microscope, including optics and incubation chamber settings.
L207: Statistical power of the study was not calculated.
L213 (and further): replace statistically « valid » by « significant ».
Results:
Fig1, Fig2: The units of the y-axis are missing. Please indicate that it is in percentage (when using relative marker area). The referee requests to keep the same y-axis (0-100%) as adjusting the graphs in this way artificially exaggerates the differences and the comparison between different compartments becomes challenging.
For Fig1C, remove the control column, as it makes no sense.
For readability, do not use 4 digits after coma but round to the unit. Indicate all points on the box-plot (« show all dots »).
L302: the authors calculated 5 decimals after 0, with only 2 decimals from measurements.
For all the figures with illustrated material: Scale-bars are missing, or unreadable (Figure5). Please place a thick contrasted bar and refer to it in the legend with the appropriate value.
Fig 3-5: please add quantifications as in Fig1-2.
For MMPs, specify whether the study investigates the pro-form or active form (see Fig 5), and consider discussing the regulation of MMPs, including activation, inhibition, and recapture, as well as the role of TIMPs.
Discussion:
Include necessary references (L340, L349,...)
L344: Correct "mRNA".
L408 and further: Avoid repeating results or introducing new ones in this section.
Discuss the limitations of the study and provide perspectives.
Conclusion:
The conclusion should not introduce concepts that were not previously introduced or discussed.
General Comments on Presentation:
Address inconsistencies in writing and grammar.
Include the country of manufacturer for relevant materials.
Figures must be revised (see above).
Global recommendation: The manuscript requires substantial revision before resubmission. Here are the key points that need to be addressed:
1. Methods: The methods section needs to provide further detail. It should include a more comprehensive description of the experimental procedures and techniques used in the study.
2. Result presentation: The presentation of the results needs to be revised. There is place for a clearer narrative, too.
3. Novelty: The authors should aim to enhance the novelty of the manuscript. Previous research has already explored the expression of kisspeptin-1 and its cognate receptor in endometriosis, which limits the novelty of this study in that regard. It is recommended to explore additional aspects related to the interplay between Kisspeptins/MMPs in the context of endometriosis. For example, (a) investigating the role of hyperestrogenism, (b) exploring the contribution of collective-cell migration, or (c) analyzing samples derived from patients undergoing kisspeptin therapies or (d) testing these drugs in the current settings could provide new insights.
Addressing these points would help to improve the manuscript and increase its scientific rigour and impact.
Comments on the Quality of English Language
The abstract and body of the text need to be revised for grammar and improved for accurate scientific English. The use of conventions also needs to be respected.
Author Response
We thank the distinguished Reviewer for the overall positive assessment of the article and very important comments. We have made all the necessary corrections to the text of the article. All corrections are marked by gray color.
Abstract.
We revised the abstract and added the information about MMPs in it.
Introduction.
The presentation of clinical manifestations of endometriosis is necessary for understanding mechanisms of this disease pathogenesis. In this regard, we kindly ask the respected reviewer to leave this fragment in the text.
We added the information about expression of kisspeptines in the uterus and the role of oestrogens in kisspeptin regulation and gave two appropriate references.
It was previously shown that pregnancy increased the uterine expression of Kiss1 and Kiss1R. These molecules can regulate INFy, MIF, VEGF, IL10, and TNF synthesis. Higher placental expression of Kiss1R mRNA occurred at the late pregnancy, while the expression of Kiss1 was higher in the mid of the pregnancy. We can suggest functional links between Kiss1 and KISS1R in placental angiogenesis and immunology [Santos, L.C.; Dos Anjos Cordeiro, J.M.; da Silva Santana, L.; Santos, B.R.; Barbosa, E.M.; da Silva, T.Q.M.; Correa, J.M.X.; Niella, R.V.; Lavor, M.S.L.; da Silva, E.B.; de Melo Ocarino, N., Serakides. R.; Silva, J.F. Kisspeptin/Kiss1r system and angiogenic and immunological mediators at the maternal-fetal interface of domestic cats. Biol Reprod. 2021, 105(1), 217-231]. The hypothalamic-pituitary-gonadal (HPG) axis play the key role in reproduction, in particular, in the synthesis of estrogens. It was shown that kisspeptines play an important role in regulation of processes of female follicle development, oocyte maturation, and ovulation through the HPG axis [Xie, Q.; Kang, Y.; Zhang, C.; Xie, Y.; Wang, C.; Liu, J.; Yu, C.; Zhao, H.; Huang, D. The Role of Kisspeptin in the Control of the Hypothalamic-Pituitary-Gonadal Axis and Reproduction. Front Endocrinol (Lausanne). 2022, 13, 925206].
MMP2 and MMP9 are the key factors of endometriosis progression and pathogenesis. However, other MMPs also can take part in endometriosis development. MMP7 promoted epithelial-mesenchymal transition in ovarian endometriosis. MMP14 regulates function and formation of invadopodia, which controls the migration and invasion abilities of mesenchymal cells [Ke, J.; Ye, J.; Li, M.; Zhu, Z. The Role of Matrix Metalloproteinases in Endometriosis: A Potential Target. Biomolecules. 2021, 11(11),1739].
The role of KISS1, KISS1R, MMPs in cell migration we investigated by Scratch Testing and Cell Invasion Assays. Of course, the continuation of this research is very interesting. But this goes beyond the scope of the tasks set out in this pilot article.
Materials and Methods:
Demographic data usually includes age, gender and place of the residence. The materials and methods of the article indicate that the study was conducted in women aged 23-38 years. We have added information about the place of residence of patients. This is Saint Petersburg, Russia.
The exclusion criteria from the study were: age over 38 years, women with concomitant diseases of the endocrine and / or reproductive system, patients whose endometriosis was combined with malignant neoplasms. We added this information in the article. The inclusion criteria from the study were: age from 23 to 38 years, women with extragenital endometriosis and healthy women without reproductive system pathology (control).
We replaced «immunohistochemistry» by «immunofluorescence» in all text in point 2.1.
We indicated the animal host corresponding to the clone and concentration of antibody in the Table 1.
The process of analyzing microphotographs was carried out according to the following algorithm. 1. Open a microphotograph in the ImageJ program. 2. Set the pixel/micron ratio. 3. Convert the image to RGB stack (the program only works with black and white images) 4. Select the studied parameters: expression area, average value, minimum, maximum. 5. Remove the glandular gaps from the image processing field. 6. Using Threshold to apply masks (examples of microphotographs with masks are shown in the figure) 7. Save the data in the excel table (at this stage it is necessary to manually delete artifacts, if they are on microphotographs). We added this information and two microphotography with masks in the article.
The references Paltsev et al., 2016 and Carbone et al., 2020 describe examples of the use area expression as an indicator to evaluate the synthesis of various signaling molecules in organs and tissues. We used a similar approach in our work.
Spectrophotometric analysis was used to assess the purity of RNA. The ratio of optical density at the wavelengths of 260 and 280 nm (A 260/280) was about 2. We added this information in the article.
We replaced «quality» by «integrity» in point 2.2 in Methodology.
RNA integrity number (RIN) in our investigation was 8. This is acceptable for data analysis as wrote respected reviewer and described in the article «Method Optimization for Extracting High-Quality RNA From the Human Pancreas Tissue”. We added this information in the article.
Yes, it was previously demonstrated the validity GAPDH as one of the best housekeeping genes for the normalization of PCR in human cell. It described in many articles, for example.
https://pubmed.ncbi.nlm.nih.gov/34861601/
https://pubmed.ncbi.nlm.nih.gov/34879096/
https://pubmed.ncbi.nlm.nih.gov/17430924/
We added this information in the article and gave the reference https://pubmed.ncbi.nlm.nih.gov/34861601/
We amended «ΔΔCq» on «2-ΔΔCq».
We used Olympus CKX53 inverted microscope with the integrated phase contrast system, 40x. We added this information in the point 2.3 in Methodology.
Statistical power (sensitivity) is the probability that one or another statistical criterion will correctly reject an incorrect null hypothesis. In other words, it is the ability of the criterion to detect differences where they really exist. We used Shapiro-Wilk criterion, homogeneity of variances was evaluated with the Levene criterion.
We replaced statistically «valid» by «significant» in all article text.
Results:
We added units in the y-axis in Figures (diagrams). We removed 4 digits after coma on diagrams. We corrected and added Scale-bars in Figures (microphotographs).
We discussed in Discussion section the role of MMP-9, MMP-2 and its inhibitors in pathogenesis of endometriosis: “Active forms of MMP-2 and MMP-9 are particularly important at early stages of extragenital endometriosis development. Any changes in the balance between the activity of ММPs and their tissue inhibitors can contribute to endometriosis development. Increased of MMP-2 expression in the ectopic endometrium can indicate aggressive progression of endometriosis. KISS1 is a regulator of MMPs, capable to suppress protein transcription. Thus, we may suppose that MMP-2 and MMP-9 are less suppressed in endometriosis patients, which, in turn, can lead to active migration (cell invasion).”
Discussion:
In the beginning of Discussion, we summarized our data. On this reason there are not references in this part of Discussion. In the next text of Discussion, we gave references. We corrected “MRNA” to "mRNA". We have excluded the repetition of results description in the part of Discussion concerning the scratch test. The new wording looks like this: “Using the scratch test method, we found statistically significant differences that may be related to the ability of kisspeptin to inhibit cell migration. In endometriosis, cell migration occurred faster than in normal endometrium”.We discussed the limitations of the study and it’s perspectives: “Therefore, we may presume that KISS1, KISS1R, MMP-2, MMP-9 can be used for the diagnostic, assessment of progression of endometriosis and the effectiveness of treatment of this disease in woman from 23 to 38 years old without endocrine system pathology and oncology diseases.. The applying of these signal molecules for endometriosis diagnostic in women over 38 years old, patients with endocrine or oncology pathology requires more detailed research, because it was not included in objectives of this investigation”.
Conclusion:
We added information about KISS, HPG axis and new information about the role of MMP in the pathogenesis of endometriosis to the introduction, provided appropriate links, corrected the discussion as recommended by a respected reviewer. With this in mind, it seems to us that in conclusion all the statements are logically derived from the introduction and discussion.
General Comments on Presentation:
We corrected article text and grammar. We included the country of manufacturer for relevant materials. We corrected Figures.
Global recommendation:
We provided further detail information in the methods section. We revised the presentation of the results. We explored some aspects related to the Kisspeptins and MMPs in the context of endometriosis in the Introduction.
Reviewer 2 Report
Comments and Suggestions for Authors
The authors present a manuscript which aims to investigate the expression of kisspeptins and matrix metalloproteinases within extragenital endometriosis. Although the study has been well conducted and the manuscript has been well written, several corrections should be made to achieve better comprehension. First, the title of the manuscript should be changed as "The expression of kisspeptins and matrix metalloproteinases in extragenital endometriosis". Second, the authors shouldmention about the power limiting factors for their study. Third, the references that were published before 2008 should be replaced with newer and more up-to-date ones if possible. I think that this manuscript can be accepted for publication in Biomedicines after required corrections have been made.
Author Response
We thank the distinguished Reviewer for the overall positive assessment of the article and very important comments. We have made all the necessary corrections to the text of the article. All corrections are marked by gray color.
We changed the title of the manuscript as recommended by the reviewer.
The power limiting factors for their study are: age over 38 years, women with concomitant diseases of the endocrine and / or reproductive system, patients whose endometriosis was combined with malignant neoplasms.
We changed the references that were published before 2008 on the newer and more up-to-date:
Referee «Vigano, P.; Parazzini, F.; Somigliana, E.; Vercellini, P. Endometriosis: epidemiology and aetiological factors. Best practice & research. Clin Obstet Gynaecol 2004, 18, 177–200” to “Smolarz, B.; SzyÅ‚Å‚o, K.; Romanowicz, H. Endometriosis: Epidemiology, Classification, Pathogenesis, Treatment and Genetics (Review of Literature). Int J Mol Sci. 2021, 22(19), 10554”.
Referee “Visse, R.; Nagase, H.; Murphy, G. Structure and function of matrix metalloproteinases and TIMPs. Cardiovasc Res 2006, 69, 562–73” to “Laronha, H.; Caldeira, J. Structure and Function of Human Matrix Metalloproteinases. Cells. 2020, 9(5), 1076”.
Referee “Chung, H.W.; Lee, J.Y.; Moon, H.S.; Hur, S.E.; Park, M.H.; Wen, Y.; Polan, M. L. Matrix metalloproteinase-2, membranous type 1 matrix metalloproteinase, and tissue inhibitor of metalloproteinase-2 expression in ectopic and eutopic endometrium. Fertility and sterility 2002, 78, 787–795” to “Barbe, A.M.; Berbets, A.M.; Davydenko, I.S.; Koval, H.D.; Yuzko, V.O.; Yuzko, O.M. Expression and Significance of Matrix Metalloproteinase-2 and Matrix Metalloproteinas-9 in Endometriosis. J Med Life. 2020, 13(3), 314-320”.
Referee “Nisolle, M.; Donnez, J. Peritoneal endometriosis, ovarian endometriosis, and adenomyotic nodules of the rectovaginal septum are three different entities. Fertil Steril 1997, 68, 585-596” to “Nisolle, M.; Donnez, J. Reprint of: Peritoneal endometriosis, ovarian endometriosis, and adenomyotic nodules of the rectovaginal septum are three different entities. Fertil Steril 2019, 112 (4 Suppl1), e125-e136”.
Referee “Kotani, M.; Detheux, M.; Vandenbogaerde, A.; Communi, D.; Vanderwinden, J.M.; le Poul, E. The metastasis suppressor gene KiSS-1 encodes kisspeptins, the natural ligands of the orphan G protein-coupled receptorGPR54. J Biol Chem 2001, 276, 34631–34636” to “ Ji, K.; Ye, L.; Mason, M.D.; Jiang, W.G. The Kiss-1/Kiss-1R complex as a negative regulator of cell motility and cancer metastasis (Review). Int J Mol Med. 2013, 32(4), 747-754”.
Referee “Ria, R., Loverro, G., Vacca, A. Angiogenesis extent and expression of matrix metalloproteinase-2 and -9 agree with progression of ovarian endometriomas Eur J Clin Invest 2002, 32, 199-206” to “Liu, C.; Li, Y.; Hu, S.; Chen, Y.; Gao, L.; Liu, D.; Guo, H.; Yang, Y. Clinical significance of matrix metalloproteinase-2 in endometrial cancer: A systematic review and meta-analysis. Medicine (Baltimore). 2018, 97(29), e10994”.
Referee “Chabbert-Buffet, N.; Meduri, G.; Bouchard, P.; Spitz, I.M. Selective progesterone receptor modulators and progesterone antagonists: mechanisms of action and clinical applications Hum Reprod Update 2005, 11, 293-307” to “Nieman, L.K. Selective progesterone receptor modulators and reproductive health. Curr Opin Endocrinol Diabetes Obes. 2022, 29(4), 406-412”.
Referee ”Deryugina, E. I.; Quigley, J. P. Matrix metalloproteinases and tumor metastasis Cancer metastasis reviews 2006, 25, 9–34” to “Gonzalez-Avila, G.; Sommer, B.; Mendoza-Posada, D.A.; Ramos, C.; Garcia-Hernandez, A.A.; Falfan-Valencia, R. Matrix metalloproteinases participation in the metastatic process and their diagnostic and therapeutic applications in cancer. Crit Rev Oncol Hematol. 2019, 137, 57-83”.
Round 2
Reviewer 1 Report
Comments and Suggestions for Authors
The revised manuscript has been improved and reflects valuable work, but the authors have partially addressed the referee's comments.
1. The study's statistical robustness (power) has not been assessed, as previously requested. The sample size should be calculated and the risk of a type-II error should be adequately addressed.
2. For the same reason, the referee previously requested to display all data points on the boxplots. As a result, it is challenging to fully represent the data's diversity
3. L226: the reference provided rather supports the use of 2 HKGs using geometrical mean, and does not consider endometrial cell lines. The referee recommend reading "Validation of endogenous control reference genes for normalizing gene expression studies in endometrial carcinoma" 10.1093/molehr/gav033. Please note that GAPDH was among most variable in this report. Moreover, essential features of endometriosis are its exposure to inflammation and oxydative stress, which are known to alter the mRNA levels of GAPDH. For to these reasons, it is highly discouraged to use GAPDH as single HKG. Please consider rephrasing L228 accordingly.
4. In the current status, the authors do not indicate how the regionalized analysis (stroma vs. gland) was performed, whereas they report this result: "The analysis of the of MMP-9 area expression in the glandular component of the endometrium revealed a statistically significant increase (p<0.05) in comparison with the control group at all stages of endometriosis". The reviewer recommends differentiating between glandular and stromal expression pattern of MMP-9, as it was done here using ImageJ : "Levels of E-cadherin and β-catenin were determined with the use of a freehand selection tool measuring DAB-positive pixels and the area of glands, excluding the lumen." 10.1016/j.fertnstert.2015.05.011
Comments on the Quality of English Language
The English langage must be revised carefully. E.g., ImmunofluoreScense; analysis of the of MMP-9; Matrix metalloproteinases (MMPs) and THEIR inhibitors; MMP-2 and MMP-9 genes expression; MMPs protein synthesis; Endometrial ectopic tissue containS glands; (HPG) axis playS; KISS1-transformed cells;...
L192: for clarity "Remove the glandular gaps from the image processing field" should be rephrased as "Tissue detection and discard of glandular lumen", if ok.
L83-85: for consistency, please use "MMP-#" when referring to the enzyme.
L276-279, 299, 351: р= 0,00287 -> p=0.003
Author Response
- We calculated the sample size in order to confirm that the risk of type 2 error is acceptable for our study. The volume of samples with endometriosis is 400. The material was obtained from 10 patients. 8 slices were made for each of the 10 samples. 5 microphotographs were taken for each slice. The volume of samples for control is 350. The material was obtained from 5 patients. 12 slices were made for each of the 5 samples. 5 microphotographs were taken for each slice. The calculated value of the samples for this study is less than 350. For example, for the expression of MMP-9 in glands at stage 1 of endometriosis, this is 311. Thus, the sample sizes we estimate are sufficient to ensure that the probability of type 2 error in this study is acceptable. We have added the following information to the description of materials and methods: «For each sample obtained from patients with endometriosis, 8 histological sections were made. For each sample from the control group, 12 histological sections were made. The calculated value of the samples for this study is less than 350, and larger samples were used for the study. Thus, the sample sizes we estimate are sufficient to ensure that the probability of type 2 error in this study is acceptable».
- We tried to display all the experimental points on the graphs. This clutters up the graphs very much, and practically nothing can be viewed on them in this case. In addition, we have never seen such a presentation of data in the articles. We kindly ask the respected reviewer to leave the graphs in the presented form so that they are easily perceived by the readers of the journal.
- We have carefully read the publication recommended by a respected reviewer, "Validation of endogenous control reference genes for normalization of gene expression studies in endometrial carcinoma" 10.1093/molehr/gav033. However, it should be noted that there are talking about cancer cells (endometrial carcinoma), and the physiology of normal and cancer cells has significant differences and it is not always possible to transfer recommendations for cancer cell research to normal ones, including for endometrial cells. We agree that the reference, which we have given to the work of da Conceição Braga is not entirely appropriate. Instead, we provided a link to the article where a study of gene expression in normal endometrium using the GAPDH gene was conducted https://www.spandidos-publications.com/mmr/9/4/1355. On the recommendation of the reviewer, we reformulated the phrase as follows: "Various genes, including GAPDH, are used as housekeeping genes during PCR analysis for endometrial cells”. It marked yellow color in the article text.
- We have carefully read the article recommended by the respected reviewer Donnez O. 10.1016/j.fertnstert.2015.05.011. In the article the levels of E-cadherin and β-catenin in endometrium were assessed in the same way as we did. Perhaps, when we described the algorithm of estimating the area of expression, we did not specify that the selection of masks was carried out manually. Now we have made the necessary clarification to the methodology (the algorithm for determining the area of expression) and provided a link to the recommended article. This addition and the added link are highlighted in yellow in the article. The addition is as follows: "Points 4 and 5 were done by using a freehand selection tool measuring immunopositive pixels and the area of expression of investigated markers. The similar approach was used by Donnez O. et al in their research [29]». It marked yellow color in the article text.
- We corrected typos: ImmunofluoreScense; analysis of the of MMP-9; Matrix metalloproteinases (MMPs) and THEIR inhibitors; MMP-2 and MMP-9 genes expression; MMPs protein synthesis; Endometrial ectopic tissue containS glands; (HPG) axis playS; KISS1-transformed cells. We rephrased "Remove the glandular gaps from the image processing field" as "Tissue detection and discard of glandular lumen". We corrected in lines L83-85: "MMP#" as "MMP-#". We corrected “Ñ€=0,00287” to “p=0.003”. All correction marked yellow color in the article text.
Round 3
Reviewer 1 Report
Comments and Suggestions for Authors
The authors have addressed the comments raised by the referees.
The referee acknowledges the preliminary nature of this study, and recommends publication for later validation by the scientific community.
Point 3: Please note that the provided reference is another study that uses GAPDH as HKG without prior validation. The referee strongly recommends validating the HKGs (as done in previously shared ref) for future research to increase the chances of publication in reputable journals.
Comments on the Quality of English LanguageThe presentation has been improved. Please note for point 5: "MMP synthesis" or "The synthesis of MMPs" (not "MMPs synthesis", which is grammatically incorrect).
Author Response
We thank the esteemed reviewer for the great work spent on reviewing our article. We will definitely take into account the recommendation for a more thorough selection of HKG and will be guided by the article "Validation of endogenous control reference genes for normalizing gene expression studies in endometrial carcinoma" for continuing our research. We added a link to this work in our article. This change is highlighted in blue.
The error in the phrase " MMP synthesis" has been corrected in the Abstract and Conclusion. This change is highlighted in blue.